# Machine Learning Use for Prognostic Purposes in Multiple Sclerosis

**DOI:** 10.3390/life11020122

**Published:** 2021-02-05

**Authors:** Ruggiero Seccia, Silvia Romano, Marco Salvetti, Andrea Crisanti, Laura Palagi, Francesca Grassi

**Affiliations:** 1Department of Computer, Control and Management Engineering “Antonio Ruberti”, Sapienza University of Rome, 00185 Rome, Italy; ruggiero.seccia@uniroma1.it (R.S.); palagi@diag.uniroma1.it (L.P.); 2Department of Neurosciences, Mental Health and Sensory Organs, Sapienza University of Rome, 00189 Rome, Italy; silvia.romano@uniroma1.it (S.R.); marco.salvetti@uniroma1.it (M.S.); 3Mediterranean Neurological Institute Neuromed, 86077 Pozzilli, Italy; 4Department of Physics, Sapienza University of Rome, 00185 Rome, Italy; andrea.crisanti@uniroma1.it; 5Department of Physiology and Pharmacology, Sapienza University of Rome, 00185 Rome, Italy

**Keywords:** multiple sclerosis, machine learning, disease progression, prognostication

## Abstract

The course of multiple sclerosis begins with a relapsing-remitting phase, which evolves into a secondarily progressive form over an extremely variable period, depending on many factors, each with a subtle influence. To date, no prognostic factors or risk score have been validated to predict disease course in single individuals. This is increasingly frustrating, since several treatments can prevent relapses and slow progression, even for a long time, although the possible adverse effects are relevant, in particular for the more effective drugs. An early prediction of disease course would allow differentiation of the treatment based on the expected aggressiveness of the disease, reserving high-impact therapies for patients at greater risk. To increase prognostic capacity, approaches based on machine learning (ML) algorithms are being attempted, given the failure of other approaches. Here we review recent studies that have used clinical data, alone or with other types of data, to derive prognostic models. Several algorithms that have been used and compared are described. Although no study has proposed a clinically usable model, knowledge is building up and in the future strong tools are likely to emerge.

## 1. Introduction

In the last decade, artificial intelligence (AI) approaches—and in particular machine learning (ML)—have been increasingly applied within the medical field, with the hope of increasing diagnostic performance and efficacy of care. Currently, the leading fields appear to be those dominated by image analysis, such as pathology and radiology [1,2]. Indeed, images are readily digitized, thus becoming amenable to automated analysis. Deep neural network algorithms attain performances similar to those of well-trained radiologists in examining medical images, and AI algorithms are being approved by regulatory agencies (see [3] for a recent meta-analysis). This general frame applies also to neurology, with neuroradiology at the forefront of ML application [4].

Multiple sclerosis (MS) is among the ten neurological diseases for which ML techniques are most actively investigated [5], along with neurodegenerative diseases (Alzheimer’s and Parkinson’s diseases), emergency conditions (traumatic brain injury and stroke), epilepsy and neuropsychiatric disorders (schizophrenia, depressive disorders, attention-deficit hyperactivity disorder, autism spectrum disease). MS is also the autoimmune disease mostly addressed by ML [6]. MS typically starts with a relapsing-remitting (RR) phase that gradually turns into a secondary progressive (SP) form, during which relapse-independent progression becomes more evident [7], disability increases and the patient’s health irreversibly declines. Relapses occur randomly [8] and the duration of the RR phase is quite variable, with 50% of the patients converting to the SP form within about 30 years of disease onset [9]. With a palette of at least 12 drugs approved as disease-modifying treatments, it is nowadays possible to considerably delay the onset of the SP phase, but the benefits must be weighed carefully against the risks, which are substantial, in particular for the most effective drugs [10].

The challenge thus becomes the identification, at the disease onset, of the subjects most likely to develop an aggressive, quickly progressing form of the disease in order to start with high-impact treatments before severe disability builds up. At the same time, patients with mild forms should avoid overtreatment, with substantial benefits in terms of safety, quality of life and overall allocation of resources. To this aim, reliable early prognosis would be extremely helpful. Clinical trials would also benefit from a more focused selection of patients [11].

A long list of clinical, demographic or modifiable features associated with long-term course of the disease has been compiled and recently reviewed [10,12,13,14]: older age at onset, family history of MS, male sex, vitamin D deficiency, smoking and high body mass index are associated with faster disability progression, impairment in walking speed and depression. Among clinical features, high annualized relapse rate, particularly on-treatment relapse, and a short interval between first and second relapses are important predictors of long-term disability outcomes. Polysymptomatic or motor onset, early cerebellar involvement and incomplete recovery from the first relapse have been related to poor prognosis. By contrast, sensory onset and optic neuritis have been described as favorable prognostic factors. Radiologic markers of worse prognosis include a higher number of T2 lesions at baseline magnetic resonance imaging (MRI), whole-brain and grey matter atrophy observed in the earliest stages, presence of spinal cord, cerebellar and brainstem lesions and increased T2 lesion number or lesion volume within the first two years. The presence of oligoclonal bands in the cerebrospinal fluid at diagnosis may also predict worse prognosis with high disability.

Although correlation between these factors and the evolution of MS is established at the population level, none of the prognostic factors or risk scores for early [15,16,17] or late [18] disease course has been validated. Thus, prediction of the clinical course of MS in individual subjects remains challenging.

In the last decade, ML approaches have been increasingly tested with regard to their capacity to provide support to patient counseling, prognosis and therapy.

## 2. What Can Be Gained from Machine Learning

Machine learning is a data-driven approach which covers a very broad set of methods. Indeed, learning machines aim to extract possibly complex relations among available data and generate predictions for an event, yielding (or not, depending on the approach used) information on the underlying processes, or on the features most relevant to the result obtained [19]. Some commonly used ML methods are explained in Section 5. Depending on the quantity and nature of available data and on the relative human-to-machine decision-making effort, ML techniques have different nuances of interpretability. The spectrum ranges from easily interpretable models, such as linear or logistic regression, linear support vector machines (SVMs) and decision trees, to fully machine-guided (obscure) nonlinear models, such as those obtained by neural networks, nonlinear SVMs, or random forest and even more complex algorithms. In these last cases, ML methods allow for an examination of the data that does not require human-derived hypotheses on how input variables combine to produce the output and a model can be constructed using a fully data-driven approach. This approach is particularly useful in the presence of complex, nonlinear interactions among the data, when nonparametric classifiers are preferable. However, even when the important features are disclosed, their significance in the natural process investigated is not always automatically explained and humans remain in charge in understanding what the features really mean.

Machine learning techniques can be classified in three main branches: unsupervised, supervised and reinforcement learning (Figure 1).

Unsupervised learning involves a dataset made up of unlabeled vectors of features and it is used to extract patterns from data and/or group them, by some notion of shared commonality, when there is no a priori knowledge on the underlying structure of the data. This technique is most commonly used to detect clusters within groups or inspect the structural relations between data.

In reinforcement learning the algorithm tries to solve an assigned problem by a trial and error procedure, using not only fixed data but also by interacting with the environment and receiving feedback, measured by rewards or penalties, for the actions it performs. It is mainly used to make an agent learn to interact with a dynamic system (e.g., a robot balancing its weight while walking).

Finally, in supervised learning, machines are trained on a collection of labeled examples where the outcome is known and is used in the form of labels to drive the training procedure. The learning algorithm seeks a mapping from the input features to label outputs that can generalize to new examples. Supervised learning is typically used to obtain predictions for classification problems, e.g., for binary outcomes, such as “healthy vs. diseased”, or, for regression problems, to get predictions on continuous outcomes. The pipeline used is summarized in Figure 2.

Machines are first trained on the training set, and then tuned on a validation set to assess the quality of the selected model and to detect the occurrence of the overfitting phenomenon during the training phase. Overfitting means that the machine constructs a very complex model that performs very well on the training set but has poor performance on new data and thus is useless. Finally, the reliability of the predictions (generalization performance) is assessed on a third set of data, the test set, comprising data that are not passed to the training algorithm and thus representing data never seen before, which implies that the model must be able to generalize. The performance of the model is defined by comparing outcomes predicted by the ML classifier to the outcome determined by clinical gold standard procedures, using indicators that depend on the type of task (classification or regression). These steps may need to be iterated before a model is ready for deployment.

When the available data are not enough to be divided into training, validation and test sets of adequate sizes, a k-fold cross-validation procedure is adopted, which does not require the definition of a static validation set but dynamically defines it, splitting the training set into k subsets and iteratively using k-1 subsets to train the model and the k-th subset for validation. This process leads to a complex double-phase procedure for defining the definitive ML model [20,21].

To evaluate the performance of the model, different indicators are used, depending on the aims of the study. For regression problems, three metrics are among the most popular performance indicators: mean square error (MSE) or root MSE (RMSE), mean absolute error (MAE) and mean absolute percentage error (MAPE). In classification tasks, instead, several indicators are calculated starting from the confusion matrix, in which correct and erroneous predictions are stated by reporting the actual values by row and the predicted ones by column (Figure 3).

Most of the datasets used in ML applications for clinical problems include a number of patients (or “records”), with several clinical or instrumental parameters (“features”) considered for each one of them. ML classifiers are designed to work on large sets of data in which, according to several authors, for instance [22,23,24], the number of records should be at least 5 to 20 times the number of features. If there is not enough data the learning algorithms might either develop overly simple models, unable to capture the complexity of the process (underfitting) or solely learn the patterns specific to the few records in the training set (overfitting). In both cases ML models do not generalize to new data, thus leading to large errors on the test set, limiting the usefulness of the study. Further explanations of these terms have been recently provided, with examples, in the field of neurology [25], while some widely-used methods are explained in Section 5.

## 3. Machine Learning and Multiple Sclerosis

In the specific case of MS, many factors, modifiable or not, subtly influence disease development and progression, and even large correlative studies have yielded to weak results [10]. Since good prognostic indicators have remained elusive for decades, it is time to explore the potentiality of a data-driven ML analysis.

In the field of MS, ML approaches have often focused on automatic examination of MRI images to classify disease at the time of onset or to predict evolution of clinically isolated forms, following the flourishing stream of image analysis. Recent reviews have summarized the state of the art [26,27] and huge efforts to identify the MRI determinants of progression are ongoing (for instance, the collaborative network awards offered by the International Progressive MS Alliance). However, although studies on automatic analysis of brain MRI scans in MS date back to 1998 [28], the approach still remains outside clinical practice. Moreover, consistent use of MRI data in studies using real-world data may require centralized analysis of images at specialized sites [29].

Nonetheless, as in most other diseases, scans of the central nervous system provide only a part of the clinical information on MS patients and must be placed into a context. Being a complex disease, MS outcome depends on diverse factors, such as genome, microbiota, lifestyle and living place. Gene expression has been analyzed using machine learning approaches in at least two studies [30,31]. In line with the idea that polygenic scores are important to understand complex diseases [32], a genetic model of MS severity has been obtained that identifies linear and complex nonlinear effects between alleles by means of a random forest ML technique [31]. It will be interesting to see how predictions from gene expression match or complement those derived from other types of data. ML analysis of evoked motor potentials also shows the potential to predict disease outcome [33].

### 3.1. Clinical Data

Clinical data, stored in health records, are commonly available, often in digital form and suitable for automated analysis, and a large amount of longitudinal data can be derived from clinical registers, which are being actively implemented for MS [34,35]. This represents important added value for the formulation of predictions on a condition evolving over decades. Practical experience shows that clinical data have a good predictive value with regard to long-term (over 15 years) outcomes in MS [36]. In studies on ML applications, when the relative importance of clinical and imaging data was evaluated, the former performed well. Here we summarize published ML studies that have used clinical data for prediction. Inspection of the results of a PubMed search with keywords “multiple sclerosis” AND (“machine learning” OR “artificial intelligence” OR “neural network”) retrieved 286 studies, of which eight used clinical data to derive predictions on the course of MS in individual patients. The papers are listed in Table 1, together with one work identified among the references cited by another review [25]. No additional studies were found among the first 100 results (ranked by relevance) of an identical search performed on Google Scholar, which yielded 3980 results.

We can divide published works into two main groups: those in which MRI data are included in all records (typically prospective studies), and those in which only a subset of records contains imaging data. This occurs in retrospective studies using data collected in the routine clinical practice, as usually patients undergo more clinical assessments than MRI scanning, so that more clinical than imaging data exist. Studies from the first group [38,39] indicate that clinical data have discriminative values for prognosis, even when used together with the results of MRI images analyzed by convolutional neural networks to identify latent lesion pattern features [39]. Examples from the second group of studies show that adding features related to results of MRI exams can even lead to a decreased model performance, due to the reduction in the number of records used and thus in the records/feature ratio [40,42]. So, basically, the relevance of clinical vs. imaging data is determined by the amount of data available.

Another point highlighted by the abovementioned studies is the value of historical data series to predict future worsening. All the papers listed in Table 1 used data collected at baseline to predict future outcome. Two studies [42,43] included in the database data collected at multiple visits and used the data related to one visit at a time. This “visit-oriented” approach is valuable if one thinks that the primary goal of the whole field of study is the identification of people at risk of rapid disease progression as soon as possible after the first clinical episode.

However, some studies [40,42,44,45] have used as input for the classifiers the changes of clinical values over various time intervals, with effects on prediction outcome. In particular, analyzing patient clinical history using recurrent neural networks, such as long short-term memory (LSTM) networks [42], improved reliability for predictions over long time intervals. There was a substantial decrease in the amount of data available for this approach, since all the records related to a patient were collated into a single time series. Nevertheless, the positive predictive value markedly increased, unfortunately at the expense of a reduced sensitivity; that is, the rate of correct identification of worsening patients. This is not unexpected as, in unbalanced datasets, misclassification is high for the less-represented class [46] and, in the population considered, very few records were related to worsening patients but no corrective procedure to mitigate class imbalance was applied [42]. In other instances, including the “visit-oriented” approach in the same study, the imbalance problem was addressed, as discussed in Section 4.2.

The most widely used ML models were logistic regression [37,39,40,41,43,44,45], linear [38,40,41,43,44,45] and nonlinear [42] Support vector machines, decision trees [37,41,45], random forest [39,41,42,44], boosting methods [41,42,44] and various types of neural networks [37,39,42] (see Section 5 for an explanation). Almost all studies used more than one ML model for their prediction and some compared their performances [37,39,40,41,42,44,45]. Logistic regression (LogR) was tested in seven out of the nine papers and it never appeared as the best performing method. Linear support vector machines (L-SVMs) appeared in six papers, obtaining the best performance in three of them, when compared with LogR, k-nearest neighbors (KNN) or decision trees (DTs). Neural networks (NNs) or convolutional neural networks (CNNs) performed fairly well: for instance, in [39] the values found for accuracy, sensitivity and area under the receiver-operated curve (AUC) (Table 1) were 9%, 17% and 9% larger than found with LogR. When used [41,42,44], ensemble models (e.g., random forest) and neural networks [37] performed better than the others. This result is reasonable, given the greater capability of these models to capture complex, nonlinear relations among data and thus to generalize. Altogether, these papers show that ML approaches can be used to address the prediction of MS in individual patients.

Last but not least, some studies [37,38,39,40,41,44,45] have tried to extract the most informative features for each ML model, with both clinical and computational aims. For the former, there is the hope of shedding light on the contributions of different clinical features to the disease course. For the latter, it is known that ML performs better when the number of features is adequate to the number of records available for the study [22,23,24], and that elimination of noninformative features can reduce the effect of noise and the risk of obtaining complex models that overfit data. The results about feature extraction are quite scattered (Table 1). Indeed, even within a single study, the most relevant features depend on the ML model used, the source of the database, and whether the focus is to identify “worsening” or “non-worsening” patients, i.e., whether to maximize positive or negative predictive value, respectively. Although presently not really informative, this type of analysis is nevertheless extremely important and one can hope that in the future it will yield valuable information [47]. As a matter of fact, ML approaches applied to neuroimaging already provide good insight into latent features contained in images [4].

### 3.2. Patient-Derived Data

In addition to the well-established use of imaging and clinical data, other types of information have been considered to predict MS course. One interesting approach is the use of patient reported outcomes or of data collected via smartphones or wearable devices (see, for example, [43,48]; for review, see also [35]). In this way, large quantities of data can be gathered, while accesses to the outpatient facility can be reduced, with considerable savings in time and effort for the most disabled patients. These studies are mostly aimed at assessing the feasibility of the approach, yet they are valuable as they pave the way for massive use of patient-derived data that can eventually lead to defining how lifestyle impacts on disease course.

## 4. Problems and Future Hope

Data represent the heart of ML methods. As underlined by the paradigm “garbage in garbage out”, the quality of the data heavily affects the performance of ML models, with reliable models obtained only with reliable data. The quality of a dataset can be assessed according to different parameters, which range from the quantity of data to the reliability of them. When ML is applied to the MS field, data are usually characterized by some of the most challenging aspects, hindering the performance of ML methods and preventing their applications as supporting tools for physicians in defining prognosis and choosing therapy.

### 4.1. Amount of Data

In studies using “real-world” data, the number of patients involved and the amount of information gathered for each patient are often insufficient to adequately train ML models. The scarcity of data is typical of prospective studies, in which a limited number of people (diseased and/or healthy controls) are enrolled. In retrospective studies, more records are available, but they can be incomplete (see below). In both settings, data must be handled in such a way as to preserve patients’ privacy, as mandated by the Declaration of Helsinki and other rules, such as the European General Data Protection Regulation (GDPR, valid in the European Union), or their counterparts in non-EU countries (for comparisons, see [49]). Apart from obvious precautions, such as patient names replaced by unrelated identifiers or date of birth replaced by age at disease onset, other issues must be addressed. For instance, in a database from a single center, rare pediatric cases can be readily identified from the low age at disease onset, so that anyone knowing the subjects will have access to their clinical data. Along the same line of reasoning, the place of origin or ethnicity (important pieces of information in MS) might lead to identification of patients from abroad/minority groups (see [50]). Thus, privacy protection requires that some potentially useful data are eliminated; a trade-off between formal guarantees of patients’ privacy and accuracy of results is necessary.

As indicated in Table 1, the number of records used in the ML procedure rarely exceeds 2000. The scarcity of data is particularly cogent in the case of supervised learning, as it requires data to be split into train and test sets, and possibly a validation set. In this respect, all these splitting procedures stick to a stringent requirement: when more records relate to the same patient, they must be included in one set only (the so-called “leave one group out” cross-validation). In fact, the relatively small size of the datasets leads ML models to identify specific patients rather than patterns in the global population. This lack of information represents a bottleneck in the development of reliable models, which need large amounts of data to properly extract patterns from them.

### 4.2. Class Imbalance

When dealing with classification problems, databases that contain very different numbers of records for the positive and negative classes (such as “healthy” vs. diseased” or “worsening vs. non-worsening” in the case of MS patients) pose a problem. In fact, if ML algorithms are trained on highly unbalanced datasets, the ML model can be biased towards the majority class. This is a typical problem in MS research, as disease progression has a slow course and, therefore, datasets often contain many more records related to the “non worsening” than to the “worsening” class. If the problem is not correctly addressed, the resulting biased models tend to overlook “worsening” patients, missing altogether the final goal of the study. To reduce this biasing effect, balanced training sets are built using only a subsample of the majority class, then bootstrap aggregation (or ”bagging”) or balanced-ensemble methods are usually applied (see Section 5.5). Several approaches are used to construct adequate training sets, such as resampling techniques like the Synthetic Minority Oversampling Technique (SMOTE; [51]). In the studies discussed above, class imbalance was addressed by training the machines on datasets with the same number of records for the worsening and non worsening classes [38,40,41,42] or using cost-sensitive learning [44].

### 4.3. Missing or Incorrect Data

In clinical databases, data are usually inserted manually by an operator. This operation can be easily affected by clerical errors [52] that lead to bugs in the data (for instance, different sex indications in subsequent records related to the same patient). Moreover, fields relating to nonrelevant information are often skipped, leading to the presence of missing values. These issues further hinder the effectiveness of ML techniques, which are affected by the presence of misleading information. Data preprocessing techniques must be implemented to reduce the noise introduced by errors and impute the missing data. It is important to note that these approaches are useful to eliminate subsets of poor-quality data from otherwise "good" databases. No present or future technique, however advanced, will ever be able to increase the quality of a poor database: the primary data must be properly recorded in order to be useful. Thus, clinicians creating datasets must be strongly committed to prepare data adequate for ML applications.

### 4.4. Generalizability

The clinical status of the patient is rated also on the basis of subjective considerations by the visiting neurologist, leading to inter-rater (but also intra-rater) variability that adds a further source of noise and misinformation in the data. The Expanded Disability Status Scale (EDSS) itself is assigned by attending neurologists and is subject to personal opinion, although variability is minimized in large MS centers, where experienced clinicians adhere to internationally recognized standards. The physical status of the patients can be assessed by more objective parameters, such as the Timed 25-Foot Walk test or the Two-Minute Walk Test, or compound scales [17,18]. However, these indicators do not provide a comprehensive picture, given that MS affects not only motor performance but also sensory and cognitive functions, often comprising fatigue and pain, which are by definition subjective experiences. Several MS experts suggest the use of indicators based on the evaluation of cognitive functions, which are potentially very interesting in the ML context as cognitive performance can be assessed in an automatic and quite objective manner, and this has shown predictive value in preliminary studies [53,54,55].

As a final aspect, the generalizability of ML models to different settings from the one where they were trained is a further limiting factor [56]. This is well-exemplified by the case of MRI imaging. Usually, ML models are trained on a dataset obtained by means of a specific instrument and generalize poorly to images collected using other equipment. A clear example of reduced classification accuracy upon generalization was recently provided in a study aimed at predicting the one-year outcome of Clinically Isolated Syndrome using images collected in several European centers [57]. However, efforts to solve this problem are underway (see, for instance, [58]). Similarly, clinical or patient-reported features considered for developing a ML model might reflect local cultural influences and therefore encode some “hidden” patient characteristics. When applying the same model on patients from different regions (and thus with a different culture), the same feature might not encode the same characteristic and the model performance can decrease as well. Thus, the development at a wide scale of ML models for MS data (and, more generally, for healthcare data) follows a difficult path. Two alternative approaches are possible to improve generalization: the definition of very large and comprehensive datasets that include individual variability and the development of computational frameworks that are then trained on regional datasets, taking into account local variability [56].

### 4.5. Data Fusion

The development and course of MS depend on so many factors that it is difficult to envisage a single model capturing all the complexities of the condition. We can therefore reasonably suppose that a successful approach will be one that puts the pieces together, fusing data obtained via different approaches. There are several ways of fusing data together, as recently rigorously reviewed for deep learning [59], and we wish to comment on some perspectives offered by some of these methods. Several studies related to MS have attempted data fusion. Some [38,39,40,41,44,60] represent examples of “early fusion”, intermediate between type I and type II fusion (as defined in [59], meaning that some original features, such as clinical data (type I fusion) are joined with extracted features, for instance MRI results, (type II fusion) in the records used for ML analysis. This fusion paradigm has the advantage of being flexible and implementable at any level of specialization of the center providing the data, thus favoring the spread of ML applications to “real world” settings. The “late fusion” modality, in which different types of data are processed separately and their outputs aggregated (as done, for instance, by [37]) offers interesting opportunities. Each model acts as an independent agent that captures specific aspects of the problem, embedded in different sets of data. The aggregation of predictions by independent agents often enhances their global validity (see [61] and references therein). A further point of interest is related to the confidence of the rater in making the predictions. The correctness of human medical predictions correlates well with the confidence of the predictor and influences the outcome of aggregation of independent predictions [62]. An index of confidence can be introduced for ML models too [42] and it will be interesting to see how it works in fusing the predictions of ML models.

### 4.6. Explainable Machine Learning

As we mentioned, the spectrum of data-driven machines ranges from fully human-guided models to fully machine-guided ones. When speaking about ML, one is usually referring to a predictive modeling perspective that uses general-purpose learning algorithms, such as deep networks and nonlinear support vector machines. Although powerful and effective in detecting relations between input and output, these approaches are still regarded as black-box models, in which the mechanism linking the input to the output is difficult to understand. This aspect is particularly disliked by physicians, who are often interested in explanatory or descriptive models [63]. This might represent a limiting factor for the adoption of more complex ML models. Until physicians are able to understand why models return specific outcomes, they will not use them as decision support systems [64]. Thus, unraveling the mechanisms that make a model opt for a specific outcome would be a milestone step to encourage the adoption of these methods.

In recent years, many steps have been taken in order to unravel the black-box approach of many complex machine learning approaches. Several techniques have been developed to determine the importance of features and how they contribute to specific predictions, such as Local Interpretable Model-agnostic Explanations (LIME) [65], DeepLift [66] and, recently, the novel Shapley additive explanations method (SHAP) [67]. To the best of the authors’ knowledge, however, these explainable ML techniques have not been investigated in-depth to analyze the influence of features in deep models for MS applications. Thus, an interesting field of research remains open for future development and interesting collaboration between clinicians and computer scientists.

## 5. Brief Description of Commonly Used Models

### 5.1. Linear and Logistic Models for Regression/Classification

Linear regression (LR) is an approach used in regression problems to model a linear relation between inputs and a scalar output. The estimation of the weights in an LR model is a well-known optimization problem for which several techniques have been developed, including for large datasets by following statistical and optimization approaches (Figure 4A, left) [68].

Logistic regression belongs to the class of generalized linear models and is used when the output has a binary distribution (i.e., binary classification problems) [69]. LogR follows an approach similar to LR, but the linear function is then squashed to obtain output within the range [0,1]. An instance is then classified as belonging to one class or the other if the value returned by the LogR is above or below a certain threshold value (usually set at 0.5). Thus, LogR naturally provides a confidence level for its predictions (Figure 4A, right).

LR and LogR are among the simplest predictive models and their simplicity and interpretability are the main reasons why they are still used in practice, in spite of their limited range of application. Moreover, the models can be easily extended by including automatic feature selection (see, e.g., the Least Absolute Shrinkage and Selection Operator (LASSO) method [70]).

### 5.2. k-Nearest Neighbors

The k-nearest neighbors algorithm is a nonparametric method used for both classification and regression tasks [71]. The main idea behind the working process of KNN is that similar instances should be “close” in the feature space. Thus, KNN predicts the output of a new sample by identifying the k points in the training data whose features are closest according to some metric distance (e.g., Euclidean distance). The output is computed as the mean of the labels of the neighbors (in the case of regression problems) or as the most frequent class (in the case of classification problems) (Figure 4B). 

KNN is a simple, easy-to-implement predictive model but it is too simple and naïve to catch highly nonlinear relations among inputs and outputs and thus has an unsatisfactory performance if compared to more sophisticated ML models. Moreover, when applied to large training sets, KNN becomes inefficient, requiring a lot of computational time to compute the k-nearest neighbors for each entry.

### 5.3. Support Vector Machine

Support vector machines were first introduced to solve classification problems [72] and later extended to regression problems as well [73]. In classification problems, given a set of linearly separable data, namely a set of points that can be divided into classes, there exist infinite hyperplanes (lines in a 2D space) separating these points. SVMs look for the hyperplane that maximizes the distance from the closest points in each of the two classes (Figure 4C).

SVMs have been extended to also consider nonlinear classification problems, using kernels, and to allow the presence of misclassified samples [74]. Overall, when dealing with classification tasks, SVMs are considered robust learners and a go-to method. The main advantage of SVMs is that mathematically they return the “best” separating hyperplanes both in the original space (linear SVM) and in a transformed space (nonlinear SVM) and there are many fast and memory-efficient algorithms to solve them [75]. The main drawback is that SVMs are not easy to interpret and are considered a black-box method by several authors, although for linear SVMs some research has been devoted to safe interpretation (for instance [76] and references therein). Moreover, they do not directly provide a probabilistic interpretation of the outcomes, although approaches to obtain confidence levels of the predictions have been proposed [77].

### 5.4. Decision Trees

Decision trees can be considered as rule-based classifiers. Given a training set, DTs define a sequence of binary rules that make it possible to correctly classify most of the samples in the training set [78]. Thus, given an instance, a DT sequentially checks whether various rules are satisfied or not and returns an outcome accordingly (Figure 4D).

Decision trees are simple and useful for interpretation. However, they typically are not competitive with more advanced supervised learning approaches in generalization performance and can easily overfit if no constraints on the maximum number of rules are considered (i.e., a maximum depth of the tree). Even though these methods are not very effective on their own, they are at the basis of many ensemble methods (such as random forests) that have proved to be very effective learning models.

### 5.5. Ensemble Methods

Ensemble methods are machine learning approaches that combine several learning models (usually DTs) in order to define a learner with a better performance. Although ensemble methods have proved to be very effective in many estimation problems, their main drawback is the lack of interpretability of the model obtained. For more details on the topic, we refer the reader to [79]. We briefly recall two main ensemble approaches: bootstrap aggregating (bagging) and boosting.

#### 5.5.1. Bagging

In bagging, B subsets are extracted from the initial training set and each subset is used to train a strong ML model (e.g., a decision tree with several rules). The final output is then obtained by averaging out the predictions returned by the B models (Figure 5, left). Random forests are bagged DTs where each DT is obtained by considering only part of the whole set of features [80]. Each DT is a strong learner, meaning it is a deep DT with many rules and thus is able to perform very well on the training set at the risk of overfitting. Random forests, by averaging several deep DTs, reduce the variance of these estimators and achieve a more robust estimator with better generalization properties. Currently, random forests represent one of the most successful examples of ensemble bagging methods.

#### 5.5.2. Boosting

In boosting, a predictive model is achieved by incrementally training a set of weak classifiers, where a weak classifier is a classifier that obtains performances slightly better than random guessing [81]. The starting weak classifier is trained on the entire dataset and the following classifiers are trained on modified versions of the starting dataset, where more importance is given to the instances misclassified by the previous models. Then, the final predictions are obtained as a weighted average of these weak learners (Figure 5, right).

Differently from bagging approaches, where models are trained in parallel on bootstrapped subsets, in boosting approaches the models need to be trained in sequence on differently weighted versions of the entire dataset. Boosting methods can easily overfit if too many classifiers are sequentially built [79]. The most famous examples of boosting methods are AdaBoost [82] (which was the first boosting method proposed) and XGBoost [83] (which is a fast boosting algorithm).

### 5.6. Neural Networks

Neural networks are inspired by the mechanisms of the human brain [84]. In a neural unit inputs are collected, weighted, summed up, nonlinearly transformed and passed to the next neuron (Figure 6, left). In NNs, a set of neural units is organized in layers where the outputs of some neurons become the inputs of neurons in the following layers (Figure 6, right). The transformation applied within each neuron and the rules according to which neurons communicate determine the type of NN model. Application-specific neural units can be implemented, such as in the image processing setting where convolutional neural networks are mostly developed, thanks to their translation invariance characteristic [85].

Deep neural networks (DNNs), NNs with several layers, are at the basis of deep learning, a branch of machine learning based on the idea of the nested hierarchy of concepts. Given a complex task, it can learn by incrementally learning more abstract representations computed in terms of less abstract ones [86]. By leveraging their layered structure, DNNs can extract more abstract features as the model becomes deeper and have proved to be effective in solving many challenging problems (see, e.g., the ImageNet competition [87]). For more information on this topic, we refer the reader to [21].

## 6. Conclusions

In conclusion, given the increasing availability of large electronic health records, the growing interest in personalized approaches to therapy, and the use of portable devices for remote diagnostics and follow-up of patients, the use of ML in healthcare is becoming unavoidable [1,34,88]. Application of AI approaches in healthcare is so widespread that a consensus statement with guidelines for correct use of AI in clinical trials has been published [89].

However, in spite of the huge number of published studies, most applications still fail to enter routine practice, even if they perform well in experiments and clinical trials (see reviews in [2,90] and, specifically with regard to neurology, in [47]). No study has identified methods pertaining to predicting the course of MS with performances usable in the clinics. Hopefully, use of more potent ML techniques and larger collections comprising several types of data will yield usable tools for clinical practice in the near future. However, the use of these resources will be boosted only once the end-users, i.e., clinicians, acquire familiarity with the tools. Adding courses on AI to the curriculum of medical students would help promote this cultural change in the future [91,92]. Such change would be further supported if it were possible to adapt ML methods to local clinical needs and routines, and for this purpose good collaboration between clinicians and computer scientists is required.

## Figures and Tables

**Figure 1 life-11-00122-f001:**
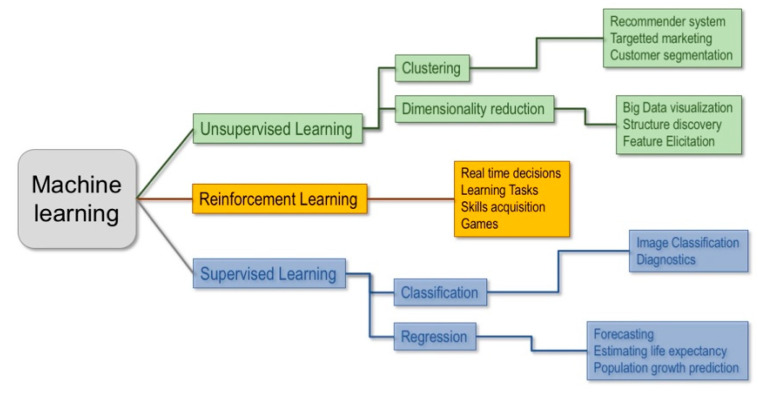
Main types of machine learning (ML) techniques and their applications.

**Figure 2 life-11-00122-f002:**
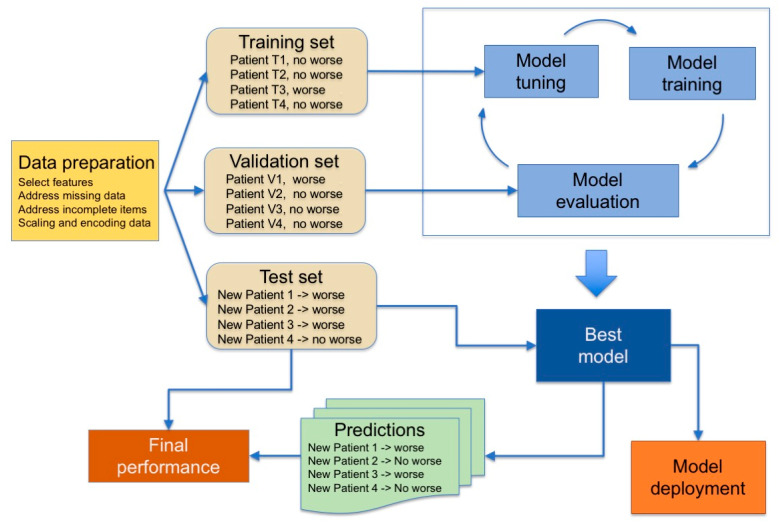
Typical steps used in supervised learning. The model tuning consists in adjusting hyperparameters according to the ML model used.

**Figure 3 life-11-00122-f003:**
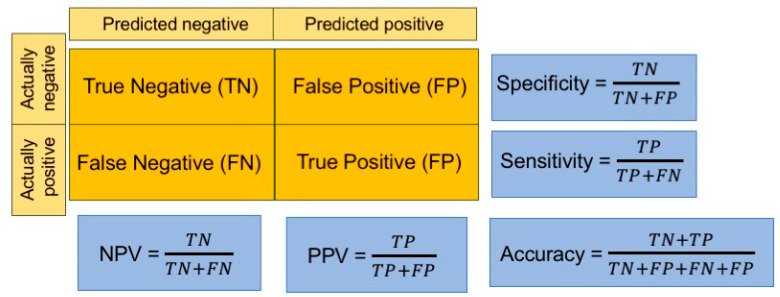
Widely used performance indicators used in classification tasks. The confusion matrix is shadowed in orange. NPV: negative predictive value, PPV: positive predictive value (also called precision). Sensitivity is also termed recall.

**Figure 4 life-11-00122-f004:**
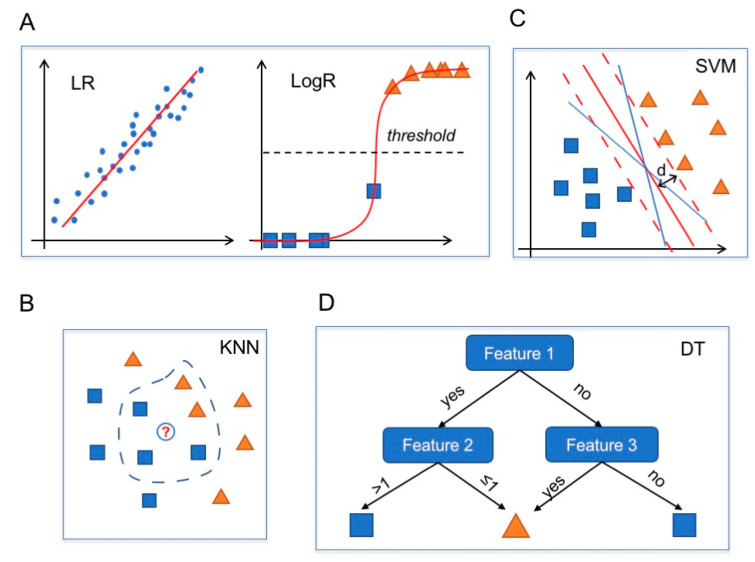
Commonly used computational models. Blue dots, blue squares and orange triangles represent data. (**A**) Linear regression (LR) model (left): the red line is the linear estimator obtained from the fit of the data. LogR model (right): the red line is the logistic model (a squashed linear regression) and the black dotted horizontal line is the threshold value. (**B**) A KNN algorithm with k = 5 classifies the white circle (new data) as a blue square, like the majority of the five nearest neighbors. (**C**) A linear SVM classifier identifies the max-margin hyperplane (red line; dashed lines indicate the boundaries; d = maximal distance) among all the possible separating hyperplanes (blue and red lines). (**D**) Decision tree of depth 2. An instance is classified as a blue square if feature 1 is “yes” and feature 2 > 1 or if both feature 1 and 3 are “no”. In all the other cases, the instance is classified as an orange triangle.

**Figure 5 life-11-00122-f005:**
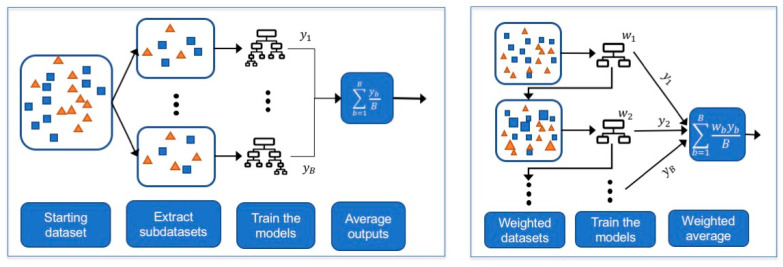
Description of bagging (left) and boosting (right) approaches. Larger squares and triangles illustrate how the instances might receive different weights when training the learners in boosting procedures.

**Figure 6 life-11-00122-f006:**
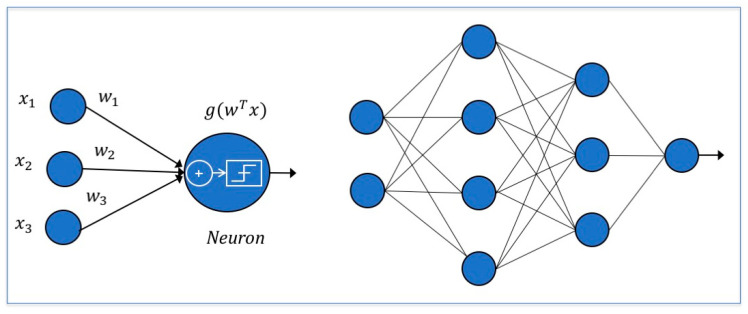
On the left is a neural unit: input features are collected, weighted, summed up and transformed by an activation function. On the right, a representation of a neural network with two inputs and one output is shown.

**Table 1 life-11-00122-t001:** Summary of the studies dealing with prognostication of multiple sclerosis (MS) course.

Reference	Subjects (Records)	Endpoint. Data Used	Model (Best Performing in Bold)	Most Informative Features for the Best Performing Model	Metrics for the Best Model
Bejarano, 2011 [37]	71 + 96 (1/patient)	ΔEDSS > 1 + EDSS range 2 years later + relapse occurrence. Clinical, MRI, MEPs	Naïve Bayes, DT, LogR, NN	EDSS, MEPs	*EDSS range:* Acc = 80%, Sens = 92%, Spec = 61% AUC = 76 % *Δ**EDSS > 1:* Acc = 75%, Sens = 87%, Spec = 52%, AUC = 74 *Relapses:* Acc = 67%, Sens = 53%, Spec = 77%, AUC = 65%
Wottschel, 2015 [38]	74 (1/patient)	CIS converts to MS in 1 or 3 years. Clinical, MRI	L-SVM	*CIS to MS at 1 year*: lesion load, type of presentation, gender *CIS to MS at 3 years:* age, EDSS at onset; lesion characteristics: count, average proton density, average distance from brain center, shortest horizontal distance from the vertical axis	*CIS to MS in 1 y:* Sens = 77%, Spec = 66% *CIS to MS in 3 y:* Sens = 60%, Spec = 66%
Yoo, 2017 [39]	140 (1/patient)	CIS converts to MS in 2 years. Clinical, MRI	LogR, RF, CNN	Not assessed	Spec = 70.4%, Sens = 78.7%, Acc = 75.0%, AUC = 74.6%
Zhao, 2017 [40]	up to 1693 (1/patient)	ΔEDSS ≥ 1.5 at 5 years. Clinical, ±MRI	LogR, L-SVM	*Non progressive cases*: EDSS at 0, 6, 12 months; disease activity at 0, 6, 12 months; race, ethnicity, family history, brain parenchymal fraction *Progressive cases*: ΔEDSS; disease activity; pyramidal function and change at 1 y; disease active at baseline, T2 lesion volume	Spec = 59%, Sens = 81%, Acc = 67%
Law, 2019 [41]	485 (1/patient)	ΔEDSS ≥ 1 at 2 years in SP MS. Clinical, MRI	Individual and ensemble LogR, L-SVM, DT, RF, ADB	EDSS, 9-Hole Peg Test, Timed 25-Foot Walk	Spec = 61% (RF), Sens = 59%, PPV = 32.1%, NPV = 82.8
Seccia, 2020 [42]	up to 1515 (up to 14,923)	RR converts to SP at 0.5 to 2 years. Clinical, ±MRI	NL-SVM, RF, ADB, KNN, CNN	Not assessed	*RR to SP at 2 y (RF):* Spec = 86.2%, Sens = 84.1%, Acc = 86.2%, PPV = 8.9% *RR to SP at 2 y (NN):* Spec = 98.5%, Sens = 67.3%, Acc = 98%, PPV =42.7%
Brichetto, 2020 [43]	810 (up to 3398)	RR converts to SP within 4 months. Clinical, patient reported outcomes	LogR, L-SVM, KNN and other linear classifiers	Not reported	Acc = 82.6%
Zhao, 2020 [44]	724 (CLIMB dataset) + 400 (EPIC dataset) (1/patient)	ΔEDSS ≥ at 5 years. Clinical, MRI	LogR, L-SVM, ensemble models (RF, boosting methods)	Value at a given time or change in 2 years of: EDSS, pyramidal function, disease category (RR, SP etc.), MRI lesions, ambulatory index, cerebellar function	*CLIMB dataset, XGBoost* Spec = 69%, Sens = 79%, Acc = 71%, AUC = 78%
Pinto, 2020 [45]	up to 187	RR to SP @ 5 years and EDSS > 3 at 6 or 10 years. Clinical, MRI	KNN, DT, LogR, L-SVM	*SP development:* EDSS, FS scores (sensory, brainstem, cerebellar and mental), CNS involvement in relapses (pyramidal tract, neuropsychological and brainstem), age at onset. *Disease severity*:EDSS, FS scores and CNS affected functions during relapses	*RR to SP:* Spec = 77%, Sens = 76%, AUC = 86%, *EDSS > 3 @ 6 y:* Spec = 81%, Sens = 84%, AUC = 89% *EDSS > 3 at 10 y*: Spec = 79%, Sens = 77%, AUC = 85%

Abbreviations used: ADB: Adaboost; Acc: accuracy; AUC: area under the receiver-operated curve; CNN: convolutional neural network; CNS: central nervous system; DT: decision tree; EDSS: Expanded Disability Status Scale; FS: functional system; KNN: k-nearest neighbors; LogR: logistic regression; MEP: motor-evoked potential; NN: neural network; RF: random forest; Sens: sensitivity; Spec: specificity; (L-, NL-)SVM: (linear, nonlinear) support vector machine; y: year(s); ±MRI: MRI data not always used; ΔEDSS: change of EDSS.

## Data Availability

No new data were created or analyzed in this study. Data sharing is not applicable to this article.

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
