# Peer review of "Machine Learning Use for Prognostic Purposes in Multiple Sclerosis"

_life, 2021, doi:10.3390/life11020122_

Round 1
Reviewer 1 Report
The manuscript has been extensively revised. I have no further comments
Author Response
Thank you
Reviewer 2 Report
The authors included very welcomed modifications, which makes it much more suitable for publication. There is still a major issue to address.
Major issue
One remaining major issue, although mostly confined to two paragraphs, is that the authors are continuing to emphasize the misguided description of machine learning as incomprehensible algorithms, which is all the more detrimental as this article is targeted at individuals who do not have enough knowledge to see the important flaws in this description. The whole paragraph from line 79 to 89, and the whole paragraph form line 359 to 365, need to be heavily edited and expanded upon. This is a major revision as it requires to make sure machine learning is correctly presented through the manuscript.
Machine learning does not provide “black box” or “relations unseen to the human mind” as a principle in the field of neuroimaging. Two of the most common machine learning models in neuroimaging to classify patients are linear support vector machines and logistic regression (mentioned by the authors themselves), and both these classes of algorithm can provide highly interpretable models. It is essential that the false dichotomy between “interpretable statistical models” and “relations unseen to the human mind from machine learning” pushed by the authors be changed. The authors on the contrary should make clinicians aware that some machine learning models can be very well interpreted. Therefore, a more valid description detailing examples of interpretable machine learning model and those which can be considered closer to black boxes (such as deep neural networks) need to be provided, with an explanation of why one is considered a black box and why the other is not. I encourage again the authors to refer again to the reference [19] they quote which shows a highly interpretable machine learning model as their main example featuring how machine learning can be used in health science. They can also refer to the article [38] they reference multiple times which clearly indicates which features contribute to the classifier performance while using SVM. Linear SVM does not have to be a black box and in certainly not in many uses in neuroimaging such as in the examples provided by the author. The sentence on line 425 for example needs to be edited as linear SVM can be highly interpretable.
Describing machine learning as a black box is wrong and highly detrimental to any efforts in properly presenting this field to clinicians. Conveying this distorted idea of ML could make clinicians even more inclined to reject ML approaches when it can often help them gain interpretable insights into the disease they are studying.
Minor issues
Line 201: “very accurate analysis of MRI images”. This is quite non-descriptive. Please indicate what you mean by “very accurate analysis of MRI images”
Line 203-204: rather the number of records decreasing, this is the ratio of records to features. To be edited.
Sentence 334-337: the generalizability is one of the most important aspect machine learning can provide insight into, so this section is essential. In addition of providing a general reference about generalizability [56], the authors could provide more to efforts trying to address it for ML applied to MS such as https://doi.org/10.1109/ISBI45749.2020.9098721
Line 372: The term “IT people” is inappropriate. Please replace it with another term, e.g. “computer scientists”
Author Response
The authors included very welcomed modifications, which makes it much more suitable for publication. There is still a major issue to address.
We thank the reviewer for appreciating our effort.
Major issue
One remaining major issue, although mostly confined to two paragraphs, is that the authors are continuing to emphasize the misguided description of machine learning as incomprehensible algorithms, which is all the more detrimental as this article is targeted at individuals who do not have enough knowledge to see the important flaws in this description. The whole paragraph from line 79 to 89, and the whole paragraph form line 359 to 365, need to be heavily edited and expanded upon. This is a major revision as it requires to make sure machine learning is correctly presented through the manuscript ...
We changed the paragraph at lines 79 to 89 (79 to 93 in the present version) as follows:
Machine learning (ML) is a data-driven approach which covers a very broad set of methods with different nuances of interpretability. Indeed, learning machines aim to extract possibly complex relations among existing data and generate predictions on an event, yielding or not (depending on the approach used) information on the underlying processes, or on the features most relevant to the result obtained [19]. However, even when the important features are disclosed, their significance in the natural process investigated is not automatically explained, and humans remain in charge for understanding what the features really mean. Depending on the quantity and nature of available data and on the relative human-to-machine decision-making effort, the spectrum of ML techniques moves from easily interpretable models, such as linear or logistic ones, to complete fully machine-guided (obscure) models such as those obtained by Deep Networks or Kernel methods or even more complex algorithms. In these last cases, ML methods allow for an examination of the data that does not require human-derived hypotheses on how input variables combine to produce the output and construct the model using a data-driven approach. This approach is particularly useful in the presence of complex, non-linear interactions among the data, especially when non-parametric classifiers are used.
And the paragraph at lines 359 to 365 (lines 367 to 377 in the present version) as follows
As we mentioned, data-driven machines spectrum ranges from fully human-guided models to fully machine-guided ones. When speaking about ML, usually one refers to a predictive modeling perspective, using general-purpose learning algorithms, such as deep networks and nonlinear support vector machines. Although powerful and effective in detecting relations between input and output, these approaches are still regarded as black-box models, in which the mechanism linking the input to the output is difficult to be understood. This aspect is particularly disliked by physicians that are often interested in explanatory or descriptive models [63]. This might represent a limiting factor to the adoption of more complex ML models. Until physicians will not be able to understand why a model returns a specific outcome, they will not use them as decision support systems [64]. Thus, unraveling the mechanisms that make a model opt for a specific outcome would be a milestone step to encourage the adoption of these methods.
Machine learning does not provide “black box” or “relations unseen to the human mind” as a principle in the field of neuroimaging. Two of the most common machine learning models in neuroimaging to classify patients are linear support vector machines and logistic regression (mentioned by the authors themselves), and both these classes of algorithm can provide highly interpretable models. It is essential that the false dichotomy between “interpretable statistical models” and “relations unseen to the human mind from machine learning” pushed by the authors be changed. The authors on the contrary should make clinicians aware that some machine learning models can be very well interpreted. Therefore, a more valid description detailing examples of interpretable machine learning model and those which can be considered closer to black boxes (such as deep neural networks) need to be provided, with an explanation of why one is considered a black box and why the other is not. I encourage again the authors to refer again to the reference [19] they quote which shows a highly interpretable machine learning model as their main example featuring how machine learning can be used in health science. They can also refer to the article [38] they reference multiple times which clearly indicates which features contribute to the classifier performance while using SVM. Linear SVM does not have to be a black box and in certainly not in many uses in neuroimaging such as in the examples provided by the author. The sentence on line 425 for example needs to be edited as linear SVM can be highly interpretable.
Line 438 (corresponding to line 425 of the previous version) has been changed
Describing machine learning as a black box is wrong and highly detrimental to any efforts in properly presenting this field to clinicians. Conveying this distorted idea of ML could make clinicians even more inclined to reject ML approaches when it can often help them gain interpretable insights into the disease they are studying.
We have tried to smooth the assertions on machine learning behaving as a black box without entering too many details. We hope that this meets the referee‘s requests.
Minor issues
Line 201: “very accurate analysis of MRI images”. This is quite non-descriptive. Please indicate what you mean by “very accurate analysis of MRI images”
The text has been changed (lines 207-208)
Line 203-204: rather the number of records decreasing, this is the ratio of records to features. To be edited.
As a matter of fact, in both papers quoted, the number of records for which the additional MRI features were available is reduced. We have however added that the ratio is strongly reduced (lines 210-211)
Sentence 334-337: the generalizability is one of the most important aspect machine learning can provide insight into, so this section is essential. In addition of providing a general reference about generalizability [56], the authors could provide more to efforts trying to address it for ML applied to MS such as https://doi.org/10.1109/ISBI45749.2020.9098721
Reference has been added (Ref 58, line 336)
Line 372: The term “IT people” is inappropriate. Please replace it with another term, e.g. “computer scientists”
Done (lines 385 and 521)
Reviewer 3 Report
This reviewer's comments were adequately addressed.
Author Response
Thank you
Round 2
Reviewer 2 Report
The manuscript better presents machine learning with the included changes.
Issue to finalize
Instead of going back and forth in discussions, the short analysis below will clearly indicate how often interpretable ML models are featured in the table you reported, how often they perform better than "black box" models, and what is the difference in performance when the "black box" models perform better. This will settle if black box models are indeed the vast majority or not in your topic of interest (summarized in your table): prediction of multiple scleroris prognosis with ML.
-----------------
Machine learning is indeed focused on predicting, but people do not exclude logistic regression, linear SVM and decision trees when they refer to ML. "Speaking about ML" does not refer to "non linear models". Importantly, many of the quoted papers on ML in multiple sclerosis described in the table, if not the majority, dealt with linear ML models. So:
- In the text, please mention which of articles in your table investigated very interpretable models such as linear SVM, logistric regression, decision trees, etc. and when these interpretable models provided the best performance (e.g. they seem to do so for [38], [40], [41], [43], [45], i.e more than half your reported results). What would be particularly useful is to mention the difference in performance between the interpretable model and the "black box" models (e.g. neural networks) when the latter performed better (e.g. [39]).
- Please refer to the step above to emphasize that common ML models used in multiple scleroris neuroimaging studies are interpretable and this is often unknown to clinicians who think that ML means black box non-interpretable models when this is clearly not the case (e.g. [38], [40], [41], [43], [45], i.e. more than half your reported results investigated interpretable models)
Once the two steps above are implemented, the article should be ready for publication
Author Response
- In the text, please mention which of articles in your table investigated very interpretable models such as linear SVM, logistric regression, decision trees, etc. and when these interpretable models provided the best performance (e.g. they seem to do so for [38], [40], [41], [43], [45], i.e more than half your reported results). What would be particularly useful is to mention the difference in performance between the interpretable model and the "black box" models (e.g. neural networks) when the latter performed better (e.g. [39])
We have named the more or less interpretable models in lines 87-89.
In Table 1, we distinguished between linear and non-linear SVM (which have a different degree of interpretability); we report in the head of the table that the "Most informative features" were related to the best performing model.
On line 289-291 we have reported the papers that used Log-R and Linear SVM and when these methods achieve the best performance.
We have added the gain in performance that NN achieve over logistic regression in [39] (lines 241-243).
- Please refer to the step above to emphasize that common ML models used in multiple scleroris neuroimaging studies are interpretable and this is often unknown to clinicians who think that ML means black box non-interpretable models when this is clearly not the case (e.g. [38], [40], [41], [43], [45], i.e. more than half your reported results investigated interpretable models)
The sentence on neuroimaging has been added (lines 258-260)
This manuscript is a resubmission of an earlier submission. The following is a list of the peer review reports and author responses from that submission.
Round 1
Reviewer 1 Report
In this review, the application of machine learning was discussed using the example of multiple sclerosis. The possible contribution and approaches were discussed. The main issues in clinical multiple sclerosis research were summarized.
The manuscript might profit from additionally addressing a few points:
Data protection and the different ways of dealing with data protection/ethics in different centers/countries should be addressed. It should also be mentioned that no advanced technology will help to solve the leading issues if the primary data are not properly recorded. The need for collaboration between physicians and medical IT should be highlighted.
The manuscript contains a few grammatical errors/typos and would profit from some language-editing.
Please check: Figure 2, step 7 is missing?
Reviewer 2 Report
Evaluation Summary
The paper aims at an audience of both clinicians and computational scientists who address research questions in machine learning applied to multiple sclerosis. This kind of review is important to highlight the motivations and challenges for multiple sclerosis prognostic for these two groups of researchers. However in this case, both major and minor inaccuracies may cause confusion or even misinformation regarding the concepts introduced, thus running counter to the laudable objective of the authors.
Public Review
The goals of the authors are to provide a review of studies predicting multiple sclerosis (MS) prognosis, with a focus on how machine learning (ML) succeeds where what the authors call “classical” statistical approaches fail. The study presents an overview of MS, highlighting the importance of early prognosis to choose appropriately a disease-modifying treatment. It also presents an overview of machine learning concepts. This is important to bring together two very different populations of researchers working on the same subject: clinicians and computer scientists.
- An important issue is the way machine learning is presented to readers unfamiliar with this field, with major and minor inaccuracies which may mislead the reader
-
- First, the article stipulates that ML is not “yielding information on the underlying processes (resulting in the events of interest)”, is “an unbiased approach” and is not “assuming any ground-truth”. All these elements are inaccurate.
Commonly used ML algorithms can provide an assessment of feature importance in the similar way as what the authors call “classical statistics”. This is clearly shown in the reference 19 brought forward by the authors themselves to describe the difference between machine learning and statistics: the figure 2 a and b of that article clearly demonstrates a very similar feature importance assessment between “classical” statistics and ML.
Regarding bias, it is also common in ML to select features manually or through the choice of a selection procedure (e.g. univariate test), all of these introducing bias. The Table 1 compiled by the authors to include all relevant ML studies show that virtually all the most informative features of these ML models were human-selected. The confusion probably arises when considering a specific class of ML models, neural networks, commonly used in the ML field when applied to fairly homogeneous data such as images for which features are automatically chosen through the network weights. This situation rarely applies however in the clinical settings typical of the subject of the article due to the importance of human-selected clinical features. The emphasis of the “unbiased” ML approach seems therefore misplaced.
Regarding the role of ground truth, it is essential in the classification settings highlighted in the article, and ML algorithms completely rely on ground truth data, both during the training phase to estimate the model parameters, and in the testing phase to evaluate the performance of the model.
2. Second, the description of the ML approach itself is inaccurate.
Unsupervised learning does not have any a priori knowledge of the label of the samples but it can certainly have a priori knowledge on the structure of the data which is for example utilized through the choice of a distance metric in clustering algorithms.
In the description of the validation set, it is not mentioned that the performance of the model is measured not only on the test but also via the validation set. The validation set is also described as useful to avoid overfitting, but this term is only defined later on. Later the authors mentioned that “the small size of the datasets allows the ML models to <<cheat>>” without explaining this is simply a case of overfitting.
Cross-validation is also missing from the main description of machine learning (section 2) and only mentioned in passing in section 4.1 despite is large importance in ML.
The authors also indicate that “ML techniques need to be trained on balanced datasets” when this is not the case. While having imbalanced datasets can often present challenges, the classes do not need to be balanced, and many techniques exist to address this situation. For example the authors mentioned that “no corrective procedure to mitigate class imbalance could be performed due to the computational burden with recurrent neural networks” when it is actually common to implement corrective procedures such as the Synthetic Minority Oversampling Technique (SMOTE) in this situation.
Also confusion is added with sentences such as “ML performs better when the number of records is balanced to the number of features” (what does “balanced” mean in this case when in most circumstances the performance increases with the number of records no matter the number of features ?) or “the most relevant features depend on (…) whether the focus is to identify <<worsening>> or <<non-worsening>> patients (although one might think that this is the same question)” (without explaining when are the situations when someone want to minimize the FPR or FNR and how this is done simply according to the choice of probability threshold for classification).
3. Third, the description of the context in which to use either ML or “classical” statistical models (if one assumes they can easily be distinguished) is wrong.
It is mentioned that “ML classifiers are designed to work on large sets of data, in which the number of records should be 10-20 times the number of features” however, as stated in the main “ML vs classical statistics” reference by the authors (ref 19) “ML methods are particularly helpful when one is dealing with 'wide data', where the number of input variables exceeds the number of subjects, in contrast to 'long data', where the number of subjects is greater than that of input variables”. Furthermore the article implies that the problem of underfitting or overfitting is specific to the “complex structure of ML models”, when exactly the same issue exist for “classical” statistical models such as linear regression where the number of predictors can directly result in under- or over-fitting (even with a few predictors, powers of these features added to the linear model with their interactions can easily cause overfitting).
B. There is also an important issue with the MS literature review on ML disease prognosis.
The MS literature review is claimed to be exhaustive while the authors admit they did not perform a systematic search. Such a claim would need to be supported at the minimum by a description on what was the protocol used to come up with their exhaustive list of MS studies addressing disease prognosis with ML.
The table 1 which is at the core of the article to list of the studies of interest seem to have serious problems. Many metrics seem to be wrong and below 50% when all these tasks are likely to be two-class problems (no precision provided by the authors on that subject). This confusion is worsened by the fact the endpoints are not well described. One has to assume that “ΔEDSS @ 5 y” is not going to be a numbered outcome (the actual value of ΔEDSS at 5 years) but instead a binary outcome obtained from an unspecified threshold T such as (“ΔEDSS@5 y > T”).
Looking at two random entries (Yoo 2017 and Zhao 2020), these are two-class studies (conversion from CIS to MS, and ΔEDSS@5 y > 1.5 respectively) and the numbers reported in Table 1 (Acc>41%, Spec>40%, Sens<28% for Yoo 2017 and Spec > 24% and Sens < 53% for Zhao 2020) do not make sense when looking at the performance of the models reported in those articles. The use of “>” and “<” signs to report metrics in that table are confusing as well.
Other important challenges of ML applications to MS are missing such as the use of EDSS which is a common endpoint (arbitrary definition, how it is measured, etc.), and others which are mentioned would benefit from being emphasized (problems of data standardization between centers and how this is addressed in research studies)
Recommendations for the Authors
The authors chose an important subject, and their objectives of highlighting the motivations and challenges of multiple sclerosis prognostic and what each group between clinicians and computer scientists should know (to collaborate in the best possible conditions) is certainly laudable.
However, I believe the entire presentation and discussion of machine learning should be changed as it is riddle with inaccuracies and sometimes even wrong statements.
The MS specific aspect of the manuscript also suffers from major issues. If the list of ML studies on the subject is to be exhaustive then the protocol to come up with this list should be described. Many of the metrics reported in Table 1, probably the numbers of most interest to the readers, seem wrong. They should all be corrected. A clear description of what the metrics correspond to should be added, e.g. to indicate they correspond to the best model. Unclear class definition should be amended: e.g. “ΔEDSS@5 y > 1.5” instead of “ΔEDSS@5 y”.
In conclusion, the manuscript would need to be entirely overhauled.
Reviewer 3 Report
This is an interesting review of the challenges and opportunities of machine learning for providing prognosis in MS.
I think readers will earn from some general explanations of the specific methodology that is elaborated in Table 1. What are the main principles of decision trees? K nearest neighbors? Multilayer perceptron neural network? Random forest? and support vector machines. These could be provided in the main text or as a supplement.